# Evaluating Influencing Factors of Audiences’ Attitudes toward Virtual Concerts: Evidence from China

**DOI:** 10.3390/bs13060478

**Published:** 2023-06-06

**Authors:** Jing Deng, Younghwan Pan

**Affiliations:** Department of Smart Experience Design, Graduate School of Techno Design, Kookmin University, Seoul 02707, Republic of Korea; dengjing0502@kookmin.ac.kr

**Keywords:** virtual concert, audiences’ attitudes, player experience, technology acceptance

## Abstract

The purpose of this study is to investigate and validate the influencing factors of audiences’ attitudes toward virtual concerts. In order to address this issue, the current study proposes a conceptual model integrating player experience factors (autonomy, relatedness, and engagement) and the technology acceptance model (perceived usefulness, perceived ease of use, and perceived enjoyment). An online questionnaire on virtual concert experiences was conducted among Chinese audiences who had attended virtual concerts previously. Structural equation modeling was then used to establish the relationships between variables. The results suggested that autonomy, relatedness, and engagement positively impacted perceived usefulness, perceived ease of use, and perceived enjoyment. Furthermore, the perceived usefulness, perceived ease of use, and perceived enjoyment were significant predictors of audiences’ attitudes. The findings of this study could provide a reference for relevant virtual entertainment providers and could also serve as a point of development and exploration for the technology acceptance model and player experience in the field of virtual concerts.

## 1. Introduction

The emergence of virtual entertainment platforms has gained popularity as a new space for music events, particularly during the COVID-19 pandemic [1,2]. The definition of virtual concerts varies in the academic literature, such as livestream concerts [3,4,5], holographic concerts [6,7,8], and concerts requiring VR hardware [9,10]. In this study, a virtual concert is defined as a musical performance in which participants are projected into a digital virtual environment in the forms of virtual avatars [11,12]. These virtual concerts are mostly game-based, e.g., those found in Fortnite, Roblox, Second Life, or Minecraft, or blockchain-based, such as those in Decentraland or The Sandbox. Those were reported as the most popular form of virtual entertainment for adults, ahead of virtual sports and virtual shopping [13].

Virtual digital technologies have transformed concert audiences’ experience [8,14]. For instance, audiences are limited by location, transportation, and consumption in the cases of traditional concerts. On the contrary, virtual entertainment platforms now enable audiences to participate in concerts without a location limitation by using digital devices such as mobile phones, computers, etc. [15]. Moreover, the integration of virtual stages, avatars, video games, virtual social networks, and other virtual technologies such as virtual environments, Music Internet of Things (IoMusT), and singer identification and field adaptation (MetaSID) have proven to enhance the audience’s experience [6,16,17]. Previous studies have discussed several factors that contribute to audiences’ experience of virtual concerts, including the sense of connection, interactivity, immersion, and a feeling of “being there”, all of which can affect audience satisfaction and attendance intentions [12,18,19,20]. Hwang & Koo highlighted the positive potential change that virtual concerts could have on the music industry [21]. Vandenberg called attention to the inability of virtual concerts to reproduce some aspects of a real one; for instance, virtual concerts fail to represent large-scale interactive rituals, such as collective quiet or dance, which are regarded among the central elements of the concert experience [22,23]. Currently, the properties of virtual concerts that influence audiences’ experience and attitudes have not been fully investigated.

Although there have been many works focusing on audiences’ experience during a virtual concert, audiences’ attitudes attract less attention in this field. Previous research, from the perspectives of gratification theory, attendance motivation, and experience economy, has suggested that audience satisfaction, attendance, and continuous immersion intention are influenced by personal identity needs, accessibility, and meaningful virtual experiences [4,6,24]. However, these studies have largely focused on the substitutability of virtual concerts for in-person concerts. With the emergence of virtual concerts as distinct options for audiences, it is crucial to better understand audiences’ attitudes toward this new form of virtual entertainment. While several studies have demonstrated the applicability of the technology acceptance model (TAM) for evaluating participants’ attitudes toward or acceptance of the virtual entertainment field, such as combining the TAM with flow experience in online games [25,26] or with self-determination theory (SDT) in gaming and online social networking [27,28], the TAM has not yet been applied to virtual concerts. Additionally, there remains limited research on the factors that influence audience experience during virtual concerts from the perspective of the player experience (PX). The growing trend of hosting virtual concerts on virtual entertainment platforms has led to the utilization of representative game elements, such as manipulative and digital avatars. These elements play a crucial role in enhancing audience engagement and fostering a sense of proximity between the audience and the performing artist [29,30]. Although the roles of “player” and “audience” have gradually merged in virtual concerts, they were quite different in previous studies [31], and PX attributes are rarely included in the research on virtual concerts. Therefore, this study aims to fill the aforementioned research gaps by assessing the audiences’ attitudes toward virtual concerts and identifying the key factors of PX that influence their attitudes in the context of the virtual environment.

The purpose of this research is to evaluate the audiences’ attitudes toward virtual concerts; a conceptual model based on the TAM is employed, with the inclusion of PX factors as antecedents. The research process involved developing hypotheses and a conceptual model that links PX and the TAM. Empirical analysis was then conducted through surveys, with data on audiences’ experiences collected from a sample of 217 individuals who had previously attended virtual concerts. Path and factor analyses were carried out using SPSS26.0 and AMOS18.0, in order to examine the structure of the relationships between the variables. Ultimately, the study yielded an extended version of the TAM that also incorporates player experiences. The significance of this study lies in its potential to help virtual concert organizers better understand the preferences of their audience on a deeper level. Furthermore, it opens up a new avenue for applying the concepts of the TAM and PX within the realm of virtual concerts.

The remainder of the article is structured as follows: Section 2 provides an overview of the literature on the TAM and PX, allowing us to develop ten hypotheses through which to frame our research model. Section 3 explains the research methodology, including the participants, data collection procedures, and statistical analysis methods. Section 4 presents the results obtained from the data analysis. Section 5 and Section 6 discuss the implications and conclusions, respectively.

## 2. Literature Review and Hypothesis Development

### 2.1. The TAM and Virtual Concerts

The TAM, initially proposed by Davis, is a well-established theoretical framework for explaining and predicting the acceptance or adoption of information technology by specific users. It postulates that individuals’ attitudes and behavioral intentions toward using information technology are influenced by two constructs: Perceived Usefulness (PU) and Perceived Ease of Use (PEOU) [32]. PU is defined as “the degree to which a person believes that using a particular system would enhance their job performance” [32], while PEOU refers to “the extent to which a technology is perceived to be free from effort.” These two constructs are determined by exogenous variables [32]. The benefits of this model have been demonstrated in numerous studies across a broad range of contexts, including human–computer interaction, e-commerce, mobile applications, education, and healthcare [33,34,35,36,37]. In the context of hedonic systems, such as those used for leisure or entertainment purposes, Perceived Enjoyment (PE) has been added to the original TAM model, which has been shown to be a stronger predictor of the attitude and intention of use than PU, particularly when it comes to technologies used for leisure or entertainment purposes [38,39,40]. PE is defined as “the extent to which the activity of using a specific system is perceived to be enjoyable in its own right, aside from any performance consequences resulting from system use” [41,42]. It is also considered integral to the media entertainment experience. The TAM has been applied in various research fields related to virtual experience, including virtual reality, augmented reality, and virtual tours [43,44,45]. It has also been employed to investigate the acceptance of entertainment-related technologies and services. For instance, the TAM has been used to examine exercisers’ and spectators’ perceptions of using social exercise platforms [46] and to identify the adoption factors influencing customers’ intentions to use in-flight entertainment and connectivity services [47]. These studies demonstrate the generality of the TAM across different research fields, as well as its applicability to emerging technologies in the realm of virtual entertainment. Since a virtual concert is primarily an entertainment experience, in this study, a revised TAM, including enjoyment variables, was selected [38]. Based on the existing literature and the defined structures, this study developed the following hypotheses:

**H1:** 
*Perceived Usefulness (PU) will have a positive influence on Attitude (ATT);*


**H2:** 
*Perceived Ease of Use (PEOU) will have a positive influence on Attitude (ATT);*


**H3:** 
*Perceived Enjoyment (PE) will have a positive influence on Attitude (ATT).*


The proposed framework evaluates audiences’ attitudes toward virtual concerts based on the constructs of PU, PEOU, and PE. However, it is important to explore the factors that influence how virtual concerts are perceived in terms of these constructs. The next section will examine various antecedents that affect audiences’ attitudes toward virtual concerts and will predict their impact on the constructs of PU, PEOU, and PE.

### 2.2. The Importance of Player Experience in Virtual Concerts

PX refers to the personal experience an individual has while, and immediately after, playing a game. It involves various dimensions, including flow, immersion, challenge, tension, competence, and emotions [48]. PX is a crucial aspect of user experience within the digital gaming context, and is examined through cognitive, emotional, and social components [49,50]. Although objective methods, such as playtesting protocols, player data clustering, and biometric indicators, are available for measuring player experience [51,52,53], players’ self-reports also provide meaningful insights into these metrics [54]; these could be collected from online self-reported data [55], but are more commonly collected from questionnaires. Hence, several scales have been developed for inquiring about players’ subjective experiences.

Various scales have been developed with which to measure PX in the context of digital games. Based on the Self-Determination Theory, the Player Experience of Need Satisfaction (PENS) [56] and the Ubisoft Perceived Experience Questionnaire (QPEU) [57] have been proposed. The Means-End Theory has led the way to the Player Experience Inventory (PXI) and mini PXI [54,58]. From the perspective of User Experience, the Game User Experience Satisfaction Scale (GUESS) [59], Game Engagement Questionnaire (GEQ) [60], and Immersive Experience Questionnaire (IEQ) [61] have been developed in order to measure PX in game environments. Additionally, for measuring gameful experiences when using non-game services, GAMEX [62] and GAMEFULQUEST [63] have been developed. Johnson validated two commonly used scales, PENS and GEQ, and provided empirical evidence supporting dimensions of flow, immersion, competence, positive affect, presence, autonomy, and relatedness [64].

The integration of gamification elements has been shown to enhance the user experience in various fields [65,66]. Consequently, the key factors of PX have been adopted in several industries, such as fitness, medical treatment, training, and e-learning [67,68,69,70]. Virtual concerts have evolved into a cohesive virtual entertainment community that seamlessly combines multiple technologies and virtual environments, presenting new possibilities for the format of virtual experiences [2,29]. Considering this context, measuring audiences’ experience in virtual concerts is essential and could be accomplished through PX measurement. Venkatesh demonstrated a positive relationship between general computer playfulness and PEOU [71], while enjoyment has been identified as a central component of PX [72], indicating the potential of PX to complement the TAM. Several studies have proved the relevance of this integration, such as Park’s exploration of the determinants of players’ attitudes toward and acceptance of mobile games using an extended TAM that includes perceived control and skill, which are derived from PX [73]. Additionally, other researchers have assessed students’ acceptance of virtual laboratory and practical work by extending the TAM to include other factors, such as perceived efficiency, playfulness, and satisfaction [74]. Based on these studies and the characteristics of virtual concerts, we propose to extract autonomy, relatedness, and engagement as key variables from PX and integrate them into the TAM as antecedents.

#### 2.2.1. Autonomy

AU is defined as “the degree to which participants felt free and perceived opportunities to engage in activities that interest them” [56]. Prior studies have highlighted the empowering role of avatars in enhancing users’ autonomy. Avatars provide users with options for identity expression, such as the ability to walk, run, fly, and even teleport [75]. Several studies in the field of player experience have demonstrated a positive correlation between AU and PE [76,77,78]. This relationship has also been verified in non-pure entertainment environments; for instance, in e-learning, AU has been linked to enjoyment of the experience [79] and perceived enjoyment of human computation games [80]. Furthermore, game autonomy has been found to be positively related to gamers’ PEOU [81]. In virtual environments, avatars enable more accurate interactions and enhance spatial awareness, which might also improve the PEOU [82]. Therefore, the following hypotheses are proposed:

**H4:** 
*Autonomy (AU) has a positive effect on Perceived Ease of Use (PEOU)*


**H5:** 
*Autonomy (AU) has a positive effect on Perceived Enjoyment (PE).*


#### 2.2.2. Relatedness

RL is a social construct that refers to the feeling of social belongingness [57]. Within the realm of online music performances, several studies have emphasized the significance of social interaction [3,19]. Swarbrick emphasized the pivotal role of social and emotional connections in motivating attendance at online concerts [83]. Moritzen further highlighted that virtual concerts are more strongly associated with social connection than streaming concerts [29].

Previous research has indicated that the PU of mobile-based social games is influenced by social relevance [73], with positive associations observed between social relevance and both PE and PU in gamified e-banking applications [84]. In addition, digital avatars have been shown to improve participants’ sense of relatedness in virtual environments, enabling the audience to experience musicians’ performances from an intimate perspective [85] and promoting social engagement in virtual environments [86]. Considering the shift toward virtual concerts replacing concerts requiring physical attendance, it is reasonable to assume that digital avatars may enhance the PEOU of virtual concerts. Thus, the following hypotheses are proposed:

**H6:** 
*Relatedness (RL) has a positive effect on Perceived Usefulness (PU);*


**H7:** 
*Relatedness (RL) has a positive effect on Perceived Ease of Use (PEOU);*


**H8:** 
*Relatedness (RL) has a positive effect on Perceived Enjoyment (PE).*


#### 2.2.3. Engagement

EG is defined as a “progression of ever-deeper engagement in game-playing” [87] or “a state of focusing one’s attention from a psychological perspective” [88]. In various scales used to measure EG, common components frequently mentioned include immersion, presence, and flow [89]. Boyle considered immersion, presence, and flow as similar in a virtual entertainment gaming experience [90]. Additionally, research has indicated that an increase in curiosity leads to enhanced immersion [91], and immersion has been found to have a strong correlation with EG [64]. Thus, in this study, we define engagement as a collection of immersion, presence, flow, and curiosity.

Previous studies have also shown that these four elements are often intertwined with each other and influence the user’s experience. For example, flow was positively correlated with music experience and immersion experience [92,93]; the sense of presence can enhance the enjoyment of game concerts [29]. In the context of the virtual world, users can gain social presence through the interaction between digital avatars [94], which could be perceived as usefulness. We therefore developed the following hypotheses.

**H9:** 
*Engagement (EG) has a positive effect on Perceived Usefulness (PU);*


**H10:** 
*Engagement (EG) has a positive effect on Perceived Enjoyment (PE).*


### 2.3. Research Model

Based on the above hypotheses, this study proposes a research model that extends the TAM with player experience antecedents in order to examine audiences’ attitudes toward virtual concerts. Figure 1 illustrates the hypothesized relationships among the variables.

## 3. Methods

### 3.1. Instruments

We conducted a thorough review of the literature related to audiences’ attitudes toward virtual concerts and developed a questionnaire as a measurement instrument. The questionnaire comprises two main parts: the TAM structure and PX factors. The TAM structure was extracted and adapted from a Revised TAM with PEOU [38,95], which was derived from the study of Davis [32] and included four constructs: PU, PEOU, PE, and ATT. PU (questions 10–13) refers to the extent to which the audience perceives that the virtual concert enhances their concert experience. PEOU (questions 14–17) refers to the degree to which the audience perceives the virtual concert procedures as simple and easy to use. PE (questions 18–20) refers to the extent to which the audience experiences pleasure and enjoyment in the virtual concert, and ATT (questions 32–34) refers to the degree of positive feelings about the virtual concert experience.

In addition to the TAM structure, the questionnaire also includes three PX factors as antecedents: AU (questions 21–23), RL (questions 24–27), and EG (questions 28–31). The AU scale was adapted from the PENS [56] and refers to the extent to which the audience feels free to engage in activities that interest them in the virtual environment. The RL scale was also adapted from the PENS [56] and refers to the extent to which the audience perceives that the virtual concert will help improve their relationships with others. Finally, the EG scale comprises four items: three questions about immersion, presence, and flow were adapted from the GEQ scale [60], and a question about curiosity was adapted from the PXI scale [54]

In total, the measurement instrument consisted of 25 items, all of which were measured on a 7-point Likert scale, ranging from 1 (strongly disagree) to 7 (strongly agree). All items used in the questionnaire were adapted from validated questionnaires and translated into Chinese.

### 3.2. Participants and Data Collection

In surveys, participation was voluntary for the sample of virtual concert audiences. The sample comes from China. We questioned participants about whether they had experienced a virtual concert before they answered the questionnaire. Those who had not experienced a virtual concert were excluded. Data were collected through an online survey platform (Questionnaire Star, a professional Chinese online survey platform). A total of 236 questionnaires were distributed, and we collected 225 questionnaires, of which 8 questionnaires were invalid after we collated the answers. Finally, a total of 217 valid questionnaires were collected. From the descriptive analysis of demographic information, 116 were males and 108 were females, 26.7% of respondents were aged between 23 and 32 years old, and most of the subjects (26.3%) were students. Of the total subjects, 25.2% of subjects had a university education, and 31.6% of the participants attended virtual concerts 1–4 times a year. Table 1 provides the demographic information of the study’s respondents.

### 3.3. Data Analysis Methods

In this study, data analysis was conducted using SPSS 26 and AMOS 18. The data analysis involved two steps: reliability and validity analysis, and hypothesis testing. First, internal consistency reliability was measured using Cronbach’s α coefficient, and composite reliability (CR) was tested using SPSS26. High values of CA and CR indicate high reliability of the tool. CA and CR values above 0.70 are recommended [96]. In order to assess the convergent validity of the constructs, we used CR values and average variance extracted (AVE) values. If the CR values and AVE values are all above 0.7 and 0.5, respectively, then the convergent validity is high [97]. Discriminant validity of the constructs was verified by analyzing the square root values of average variance extracted (AVE). The sufficiency of discriminant validity was demonstrated if all constructs were higher than the inter-construct correlations. Second, after obtaining satisfactory results in the first step, the structural model was used to test the hypotheses. The significance and size of each path coefficient were analyzed in order to test our hypotheses. Model fit indices were also assessed in order to determine the adequacy of the proposed research model.

## 4. Results

### 4.1. Measurement Tool Assessment

#### 4.1.1. Results of the Reliability and Validity Tests

Table 2 presents the results of the construct assessment. The Cronbach’s alpha values of all variables were above 0.8, indicating reliable internal consistency. The composite reliability (CR) values for all variables exceeded 0.7, indicating sufficient convergent validity [98]. Furthermore, the AVE values for all variables were above the recommended threshold of 0.5, indicating good convergent validity [98].

#### 4.1.2. The Results of Discriminant Validity Test

Table 3 shows the results of the discriminant validity test. The square root values of AVEs for all constructs are higher than the inter-construct correlations, proving sufficient discriminant validity [98].

### 4.2. Assessment of the Structural Model and the Hypotheses

#### 4.2.1. Model Fit Index

For the first step of hypothesis testing, the structural model was evaluated. Our model showed a good fit with the data (χ^2^ = 358.121, DF = 2254, CMIN/DF = 1.410, *p* = 0.00; NFI = 0.89; IFI = 0.965; CFI = 0.946; RFI = 0.895; TLI = 0.954; RMSEA = 0.039). Based on the established fit criteria [99], all model fit index values were acceptable.

#### 4.2.2. Hypothesis Testing

This study calculated the path coefficient and *p* value through bootstrapping with a sample of 217 subjects. As shown in Table 4 and Figure 2, all hypotheses are supported at a significant level of *p* < 0.05 or *p* < 0.001.

The present study’s findings lend support to Hypotheses 1, 2, and 3, positing a positive impact of PU, PEOU, and PE on ATT. The estimated coefficients for these relationships range from 0.572 to 0.611, indicating moderate to strong positive effects on ATT toward virtual concerts. Subsequent hypotheses examined the antecedents of these factors, revealing that AU has a moderate positive effect on PEOU (H4), and is a significant determinant of PE (H5). Additionally, RL in virtual concerts was positively associated with PU (H6), PEOU (H7), and PE (H8). Furthermore, EG was positively associated with PU(H9) and PE(H10).

Overall, the results suggest that PU, PEOU, PE, AU, RL, and EG are all significant factors positively impacting audiences’ attitudes toward virtual concerts. Of note, with the exception of H4 and H9, which have *p* values less than 0.05, all other hypotheses have *p* values less than 0.001.

## 5. Discussion

This study aimed to investigate audiences’ attitudes toward virtual concerts by extending the TAM with factors from PX, including AU, RL, and EG. Our key findings revealed that AU had the strongest positive impact on audiences’ attitudes toward virtual concerts, primarily through the enhancement of PE. RL emerged as the most comprehensive factor affecting audiences’ attitudes. In this section, we discuss our findings related to the TAM components and their impact on audiences’ attitudes toward virtual concerts. Furthermore, we explore the antecedents that influenced audiences’ attitudes. Theoretical and practical implications of our research are also discussed. Finally, we identify the limitations of our study and provide suggestions for future research in this area.

### 5.1. Results of the TAM and Its Antecedents

#### 5.1.1. The Results Show That PU, PEOU, and PE Had Positive Influences on Audiences’ Attitudes

Among the three components of the TAM, PE was found to have the greatest influence on ATT, which is consistent with earlier studies [38,39,100]. Recent research has also suggested that PE can significantly improve ATT toward virtual entertainment contexts [101]. Our results also demonstrated that PU was the second strongest predictor of ATT toward virtual concerts. Participants in our study found virtual concerts to be useful in terms of watching the desired performance and enhancing their effectiveness in attending a concert. This finding is consistent with Choi’s explanation that digital technology provides effective services and accurately conveys the contents of the performance in a non-contact environment [102].

Furthermore, there was a positive relationship between PEOU and ATT, although the strength of the relationship was weaker than the strength of those with PE and PU. Previous studies have suggested that PEOU increases ATT by increasing PU, rather than directly affecting ATT [103]. Overall, our study found that all three components of the TAM were significant predictors of audiences’ attitudes, which is consistent with numerous previous studies.

#### 5.1.2. The Results Show That AU, RL, and EG Affect PU, PEOU, and PE in Different Degrees

AU demonstrated the strongest positive impact on PE, followed by RL, and finally EG. Therefore, AU emerges as the most influential antecedent. This finding aligns with previous research, which suggests that satisfying the need for AU can contribute to PE [56,104]. Moreover, when the audience experiences the sense of EG, they are more likely to perceive the virtual concert as pleasant, which is consistent with Yang and Zhang’s findings [105].

RL was found to be the strongest influencing antecedent for both PU and PEOU, and was the second strongest influencing antecedent for PE. These findings highlight the importance of RL as an antecedent, considering its impact on ATT in all aspects. The positive influence of RL on PU and PEOU is consistent with previous research on virtual entertainment experiences [28,46]. However, our results contrast with the widely held assumption regarding the impact of RL on PE. Our study found that RL had a significant positive effect on PE, ranking even higher than EG. Other studies suggest that RL may have a negative impact on the audience’s experience of no-contact concerts, such as a lack of relatedness between the performer and the audience, or being disturbed by virtual audiences surrounding them [9,22,106].

In this study, the significant positive impact of RL on audience experience in virtual concerts could be explained from two perspectives. Firstly, previous studies have primarily focused on RL as the feeling of connectedness with others [107]. In contrast, this study expands the definition of RL to include the role of digital avatars in the virtual environment. Participants reported feeling a sense of closeness to certain characters in the virtual concert, indicating that the inclusion of avatars broadens the concept of RL and enhances its positive influence on the virtual concert experience. This finding is supported by recent research by Park, which demonstrated that avatars can foster emotional attachment [108]. Secondly, digital avatars and the virtual environment offer unique ways to feel close to artists and interact with other players. Participants reported feeling closer to the performers in the virtual concert, which enhances their sense of connection with the artists. This further reinforces the positive impact of RL on PU, PEOU, and PE, providing an additional explanation for the significant positive effect observed.

### 5.2. Implications

The present study offers both theoretical and practical implications. From a theoretical standpoint, this research expands the empirical TAM literature by applying the model to the context of virtual concerts. This extends the research field and highlights the relevance of TAM components, including PU, PEOU, and PE, which is consistent with previous studies. Furthermore, the integration of AU, RL, and EG into the revised TAM improves the explanatory power of the model for virtual hedonic systems. This study also introduces a new perspective, that of player experience, from which to evaluate audiences’ experience and attitudes toward virtual concerts. The incorporation of player experience elements provides key factors that directly influence audience experiences and attitude, indicating the future development of virtual concerts. By regarding concert audiences in virtual platforms as players, our research framework offers a fresh perspective and a better understanding of audience attitudes within the virtual environment. Moreover, while player experience has been extensively explored in various domains, including education, gaming, and consumer behavior, this study contributes to the literature by demonstrating that player experience elements positively influence audiences’ attitudes in the context of virtual concerts. Specifically, our findings highlight the positive influence of two factors, AU and RL, thus expanding our understanding of audience experience in virtual environments and providing new research directions from a player’s perspective when evaluating audiences’ attitudes toward virtual concerts.

From a practical point of view, this study contributes a wider and deeper understanding of the needs of virtual concert audiences to virtual entertainment platforms, and provides practical insights for the improvement of audience experience during virtual concerts in the future. Since virtual entertainment platforms are complex systems that connect to many different services, the improvement of virtual concerts can also have a considerable influence on the overall services provided by virtual entertainment platforms. First of all, we propose adopting a player-experience-centric approach in order to optimize the audience experience during virtual concerts. The platform should prioritize improving the audiences’ PE, by providing the audience with stronger AU within the virtual concert environment. For instance, the platform can expand the diversity of virtual concerts by inviting singers from different regions and languages, thereby providing the audience with a wider range of choices. Additionally, enhancing the operability of the digital avatar can offer the audience more opportunities for free activities within the virtual world.

In order to enhance the PE of virtual concerts, it is recommended that the platform focuses on improving audience relatedness, avatar customization, and performer interaction, in order to provide a more immersive experience. These improvements can be achieved by offering a wider range of options for avatar customization, including hairstyles, clothing, and facial features, as well as more extensive scene customization within the virtual environment. Additionally, the platform can enhance audience intimacy with performers and other attendees through designing interactive and communicative features for avatars. Our research has shown that RL significantly impacts the audiences’ concert experience. Therefore, implementing measures not only enhances PE, but also improves PU and PEOU for virtual concerts. In terms of improving PU, the platform can focus on enhancing audience EG through enhancing the audiovisual quality of virtual concerts and creating a more immersive virtual environment with personalized customization features. This may include incorporating character storylines and optimizing the recommendation function, based on big data, to recommend virtual concerts that match the audience members’ preferences. By taking these steps, the platform can comprehensively enhance the audience’s engagement. Finally, in order to positively impact PEOU, the platform should simplify its operational processes while maintaining positive user experiences. This can be achieved by simplifying the functional interface, removing or modifying uncommon functions, and emphasizing frequently used functions, such as recording, pausing, and returning functions during the performance. The platform should also facilitate simple operation throughout the entire concert, including audience registration, ticket ordering, and character and virtual environment control, without adding any unnecessary mental workload for the audience.

The findings of this study also have several policy implications. In recent years, numerous virtual entertainment platforms, including game platforms, have increased their cooperation with the music industry, resulting in the transformation of traditional music performances into new virtual entertainment experiences. Considering the intimate connection between the virtual world and reality, the self-representation of the virtual character within the virtual environment can significantly impact users’ real-life behavior. Therefore, ethical and privacy issues within virtual worlds necessitate the attention of policy makers. While the current virtual concert platforms do not involve violence or theft, mainstream 3D multiplayer games currently do involve such elements, and policy makers must determine whether the virtual concert environment can allow the prohibited behaviors in advance. In order to address this challenge, it is essential to identify specific influencing factors that may lead to potential issues. The influencing factors identified in this study from the perspective of player experience provide us with a good entry point. Therefore, it is necessary to consider these factors when formulating policies with which to regulate virtual concerts in the future.

### 5.3. Limitations and Future Research Directions

This study has several limitations that should be acknowledged. Firstly, the sample size and scope of the study are constrained. The participants enrolled in this study exclusively originated from China, and the virtual concerts they attended predominantly featured Western pop music performers. It is important to note that audiences’ preferences for singers and music styles are subject to variations across diverse regions and cultures, potentially exerting an influence on their attitude toward virtual concerts. Therefore, future studies could endeavor to encompass larger and more diverse samples from multiple regions. Furthermore, it would be advantageous to refine the research focus by investigating specific music genres, such as virtual classical concerts, virtual pop concerts, virtual rock & roll concerts, etc. Through the systematic categorization of audience groups within these concert types, it becomes feasible to discern distinct participation motivations and needs, ultimately yielding a comprehensive understanding of the diverse experiences and attitudes toward virtual concerts.

Secondly, the survey period of this study was relatively short, with most participants attending one or fewer virtual concerts per year. Since audience experiences are dynamic and changeable, as more people attend virtual concerts or visit virtual entertainment platforms, their experiences may change over time. Therefore, future research efforts may focus on the longitudinal experiences of audiences attending virtual concerts over a longer time span, which may provide more reliable and accurate results in a broader aspect.

In addition, future research efforts could explore other factors that affect audience attitudes toward and enjoyment of virtual concerts. For example, researchers could investigate the role of social influence, such as peer recommendations and social norms, in shaping audiences’ attitudes toward virtual concerts. Furthermore, studies could also explore the impact of technological advancements, such as virtual reality and augmented reality, on audience experiences of virtual concerts. Lastly, studies could also examine the roles of different types of interactions, such as social interactions and personalized recommendations, on audience experiences of virtual concerts.

## 6. Conclusions

As one of the few empirical studies that focuses on audiences’ attitudes toward virtual concerts, this research examines the issue from the perspective of player experience. The primary objective of this study was to explore and validate a measurement of the factors that influence audiences’ attitudes toward virtual concerts. We first developed a measurement instrument that combines the TAM with player experience elements, and subsequently tested six influencing factors using a questionnaire. The research findings demonstrate that, among the six factors that positively impact audiences’ attitudes toward virtual concerts, PE has the greatest influence, followed by AU, which is the most significant positive correlate of PE, and RL provides a comprehensive positive impact.

These findings are useful for virtual entertainment platforms and policy makers. On the one hand, virtual entertainment platforms can use these influencing factors as guidelines with which to develop new virtual concerts or promote their existing virtual concerts. As these influencing factors were proposed and tested by a large audience, they can be used as practical guidelines for virtual concert design. On the other hand, these findings can provide some reference comments for policy makers when designing and planning virtual concerts. As the audiences’ attitudes toward virtual concerts become increasingly positive, future policies for virtual entertainment platforms should increasingly focus on audience experiences.

## Figures and Tables

**Figure 1 behavsci-13-00478-f001:**
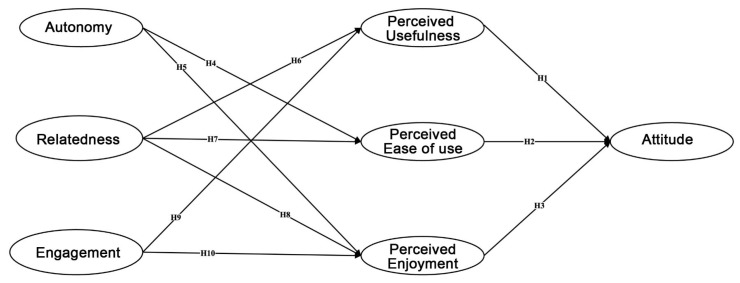
A model of the predictors of audiences’ attitudes toward virtual concerts.

**Figure 2 behavsci-13-00478-f002:**
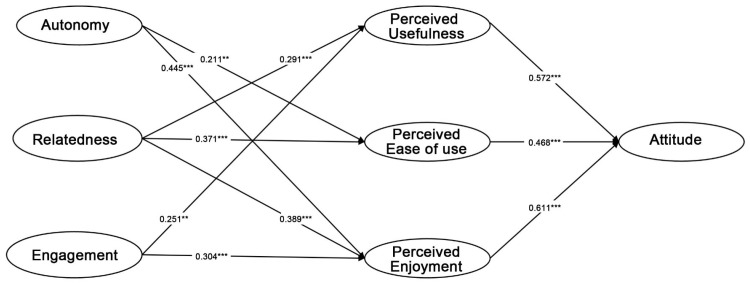
Standardized Structural Estimates and Hypotheses Tests. *** *p* < 0.001, ** *p* < 0.05.

**Table 1 behavsci-13-00478-t001:** The demographic information of participants (*n* = 217).

Variable	*n* (%)	Variable	*n* (%)
Age		Frequency of participation in offline concerts	
Under 18	13 (4.9)	Once a year or less	136 (51.1)
18–22	54 (20.3)	2–4 times a year	38 (14.3)
23–32	71 (26.7)	5–11 times a year	18 (6.8)
33–45	60 (22.6)	12 times a year or more	12 (4.5)
Over 45	19 (7.1)	Once a month or more	8 (3.0)
Gender		Once a week or more	5 (1.9)
Male	116 (43.6)	Frequency of participation in virtual concerts	
Female	101 (38)	Once a year or less	50 (18.8)
Education		1–4 times a year	84 (31.6)
High school or below	49 (18.4)	5–11 times a year	63 (23.7)
Academy	49 (18.4)	Once a month or more	16 (6.0)
Undergraduate	67 (25.2)	Once a week or more	4 (1.5)
Graduate	52 (19.5)	Frequency of playing online games	
Monthly Income		Never	11 (4.1)
Less than 2000	38 (14.3)	1–3 times a month	25 (9.4)
2001–5000	64 (24.1)	Once a month	64 (24.1)
50,001–10,000	61 (22.9)	More than once a month	81 (30.5)
10,001 and above	54 (20.3)	Everyday	36 (13.5)
Occupation			
Civil servant	26 (9.8)		
Employee	47 (17.7)		
Self-employed	28 (10.5)		
Free occupation	23 (8.6)		
Student	70 (26.3)		
Others	23 (8.6)		

**Table 2 behavsci-13-00478-t002:** The results of the construct assessment.

Variable	Mean	SD	Loading	CR	CA	AVE
Perceived Usefulness				0.839	0.882	0.567
PU1	5.06	1.678	0.702			
PU2	5.04	1.631	0.769			
PU3	5.05	1.659	0.715			
PU4	5.17	1.575	0.820			
Perceived Ease of Use				0.831	0.878	0.552
PEOU1	4.94	1.435	0.661			
PEOU 2	4.89	1.399	0.816			
PEOU 3	4.99	1.467	0.742			
PEOU 4	5.06	1.369	0.746			
Perceived Enjoyment				0.854	0.876	0.662
PE1	5.30	1.371	0.774			
PE 2	5.22	1.352	0.827			
PE 3	5.35	1.374	0.838			
Autonomy				0.752	0.823	0.504
AU1	4.93	1.472	0.696			
AU2	5.02	1.508	0.781			
AU3	5.00	1.532	0.647			
Relatedness				0.817	0.864	0.523
RL1	5.14	1.478	0.687			
RL2	5.04	1.509	0.725			
RL3	5.17	1.467	0.722			
RL4	5.14	1.513	0.770			
Engagement				0.814	0.865	0.523
EG1	5.08	1.496	0.736			
EG2	4.94	1.489	0.740			
EG3	5.14	1.473	0.761			
EG4	5.08	1.521	0.651			
Attitude				0.765	0.824	0.520
ATT1	5.38	1.399	0.713			
ATT2	5.28	1.363	0.720			
ATT3	5.51	1.385	0.732			

Note: SD = standard deviation; CR = construct reliability; CA = Cronbach’s alpha; AVE = average variance extracted.

**Table 3 behavsci-13-00478-t003:** The results of discriminant validity test.

	1	2	3	4	5	6	7
1. AU	**0.710**						
2. RL	0.351	**0.723**					
3. EG	0.445	0.389	**0.814**				
4. PU	0.341	0.291	0.419	**0.753**			
5. PEOU	0.211	0.317	0.317	0.29	**0.743**		
6. PE	0.251	0.179	0.304	0.251	0.219	**0.723**	
7. ATT	0.477	0.597	0.611	0.572	0.468	0.471	**0.721**

Note: Figures on the diagonal line (in bold) are the square roots of the average variance extracted (AVE). Off-diagonal figures show inter-construct correlations.

**Table 4 behavsci-13-00478-t004:** Standardized Structural Estimates and Hypothesis Tests. *** *p* < 0.001; ** *p* < 0.05.

Hypothesis/Path	Estimate	S.E.	C.R.	Results
Hypothesis 1: PU→ATT	0.572 ***	0.128	5.673	Supported
Hypothesis 2: PEOU→ATT	0.468 ***	0.098	4.897	Supported
Hypothesis 3: PE→ATT	0.611 ***	0.104	5.783	Supported
Hypothesis 4: AU→PEOU	0.211 **	0.097	2.578	Supported
Hypothesis 5: AU→PE	0.445 ***	0.105	4.809	Supported
Hypothesis 6: RL→PU	0.291 ***	0.12	3.524	Supported
Hypothesis 7: RL→PEOU	0.317 ***	0.099	3.741	Supported
Hypothesis 8: RL→PE	0.389 ***	0.098	4.418	Supported
Hypothesis 9: EG→PU	0.251 **	0.122	3.100	Supported
Hypothesis 10: EG→PE	0.304 ***	0.097	3.644	Supported

## Data Availability

All data generated or analyzed during this study are included in this article. The raw data are available from the corresponding author upon reasonable request.

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
