# Peer review of "Evaluating Influencing Factors of Audiences’ Attitudes toward Virtual Concerts: Evidence from China"

_behavsci, 2023, doi:10.3390/bs13060478_

Round 1

Reviewer 1 Report

The article is very interesting and inspiring.

With respect to the literature and the paper concept appropriately organized and the content is well advised. However, the use of acronyms should be presented with full explanations, for example the technology acceptance model (TAM).

I am not an expert in statistical analysis, so I cannot comment on the quality of the statistical presentation.

Author Response

Dear reviewer,

We would like to express our sincere appreciation for your careful review of our manuscript. We have taken your suggestion into consideration and made revisions regarding the use of acronyms in the resubmitted version of our manuscript. (Line 81 and line 82)

Thank you once again for your valuable time and attention. We eagerly await your response.

Yours sincerely,

Jing Deng

Reviewer 2 Report

The article is heavy on methodology, and the authors do a good job of explaining it.  I do have one suggestion as indicated to authors below. My one suggestion is that you address the type of concert, if even in a suggestion for further research.  Are audiences for a particular type of music more or less accepting of virtual concerts?

Author Response

Dear reviewer,

We greatly appreciate your valuable feedback and professional guidance. Taking your suggestions into consideration, we have made revisions to the manuscript. One notable modification is the detailed categorization of virtual concerts and audiences in the section on future research suggestions (lines 1269-1281).

Thank you very much for your attention and time. Look forward to hearing from you.

Yours sincerely,

Jing Deng

Reviewer 3 Report

In this article, the authors describe an extended model based on TAM. Due to the overlap between the behavior of virtual concert audiences and game players, the authors apply a combination of player experiences' factors to the TAM model to assess the factors influencing audience attitudes toward virtual concerts. The literature review cites a large body of literature detailing the theoretical underpinnings of the TAM model and player experiences, as well as their relevance to virtual concerts. A PX-extended TAM research model is constructed, and a total of ten research hypotheses are proposed.

A total of 225 questionnaires were collected through an online questionnaire, of which 217 were valid, and the reliability of the scale was confirmed through data analysis. The analysis also demonstrated that the research hypotheses were significant or even highly significant, indicating that the hypotheses were supported.

The main finding of the study was that AU had the strongest positive effect on audience attitudes toward virtual concerts by reinforcing PE, i.e., PE had the strongest effect on ATT, followed by PU. In addition, RL had an overall effect.

This study integrates autonomy, relatedness and participation into the modified TAM to improve the explanatory power of the model for virtual hedonic systems. It extends previous research by demonstrating the positive impact of player experience elements on audience attitudes. Finally, it is suggested that virtual concert platforms should adopt a player experience-centered approach when optimizing the audience experience by approximating the introduction of gamification elements to enhance the relevance to the audience, for example, providing customized avatar and performer interaction to provide a more immersive experience.

However, the article concludes by directly suggesting that since virtual concerts are closely related to the gaming experience, it is necessary to consider the application of anti-addiction measures and supervision in the field of virtual concerts. This section needs to be explained, as it is difficult to make a simple connection between sustainable gaming and time-limited virtual concerts, and it is suggested that this be amended or supported by the literature.

The topic of this paper is attractive and enjoyable to read, and the paper presents a considerable amount of documentary evidence, and the conclusions confirm the contribution of this paper. The study has made a considerable contribution. The methodology of the study is not new but appropriate and leads to clear results. The conclusion is complete and strong. The study also reviews a very large number of references, even more than the literature review papers, which provide a detailed discussion of the theoretical background and adequate support for the research methods and conclusions.

Finally, the article still needs some textual corrections to polish the writing to convey a clearer description.

Author Response

Dear reviewer,

We would like to express our gratitude for your thorough review and professional advice. We greatly appreciate your suggestion regarding the connection between sustainable gaming and time-limited virtual concerts. Based on your insightful feedback, we have made the necessary revisions to our manuscript. The specific corrections are outlined below:

  1. We carefully examined the relationship between the game and the virtual concert, taking into account your concerns. As a result, we have removed the policy recommendations regarding anti-addiction and have made appropriate amendments to Section 5.2 (lines 1093-1107) to ensure clarity and accuracy.

  1. We have dedicated significant effort to improving the language and overall readability of the revised manuscript.

Thank you once again for your valuable input. We believe that these modifications have strengthened the quality of our work, and we hope that the revised manuscript now meets the high standards expected for publication.

Sincerely,

Jing Deng

Reviewer 4 Report

This is an interesting and clear-presenting research.

Even though the research design, questions, hypotheses and methods are clearly stated, but H3, H5, H8, H10 are common in sense, and less valuable in discussion and conclusion.

In section 5, it maybe more clear and logical that the conclusions are listed out in another seprated section 6. 

Some spelling mistakes should be corrected, like the "Form" in line 425 should be "From".

Some sentences are too long to be read.

Author Response

Dear reviewer,

We sincerely appreciate your careful reading and valuable suggestions. Based on your insightful feedback, we have made extensive corrections to our manuscript. The detailed revisions are as follows:

1: We have condensed the discussions of H3 (lines 797-799), H5 (lines 821-823), and H10 (lines 824-826) as per your suggestion. However, we have retained the entire paragraph for explanation (lines 837-849). Since H8 differs from the conclusions of previous related studies. We have appropriately reduced the relevant description (line 827) to align with your recommendation.

2: In response to your suggestion, we have created a separate section for the conclusions, which is now listed as Section 6 (line 1354). This revision enhances the organization and clarity of our manuscript.

3: We deeply apologize for the spelling mistake you pointed out. We have corrected "Form" to "From" (line 1050).

4: To address your feedback regarding long sentences, we have made diligent efforts to polish the entire English text, ensuring that sentences are more concise and readable.

Once again, we sincerely thank you for your thorough review and guidance. We believe that these revisions have significantly improved the quality and coherence of our manuscript. We look forward to your further evaluation.

Best regards,

Jing Deng
